# Ultrasound-Mediated Blood–Brain Barrier Disruption for Drug Delivery: A Systematic Review of Protocols, Efficacy, and Safety Outcomes from Preclinical and Clinical Studies

**DOI:** 10.3390/pharmaceutics14040833

**Published:** 2022-04-11

**Authors:** Kushan Gandhi, Anita Barzegar-Fallah, Ashik Banstola, Shakila B. Rizwan, John N. J. Reynolds

**Affiliations:** 1Department of Anatomy, School of Biomedical Sciences, University of Otago, Dunedin 9016, New Zealand; ganku455@student.otago.ac.nz (K.G.); anita.fallah@postgrad.otago.ac.nz (A.B.-F.); ashik.banstola@otago.ac.nz (A.B.); 2Brain Health Research Centre, University of Otago, Dunedin 9016, New Zealand; shakila.rizwan@otago.ac.nz; 3School of Pharmacy, University of Otago, Dunedin 9016, New Zealand

**Keywords:** focused ultrasound, blood–brain barrier opening, therapeutic agent delivery, ultrasound parameters, ultrasound safety, review

## Abstract

Ultrasound-mediated blood–brain barrier (BBB) disruption has garnered focus as a method of delivering normally impenetrable drugs into the brain. Numerous studies have investigated this approach, and a diverse set of ultrasound parameters appear to influence the efficacy and safety of this approach. An understanding of these findings is essential for safe and reproducible BBB disruption, as well as in identifying the limitations and gaps for further advancement of this drug delivery approach. We aimed to collate and summarise protocols and parameters for achieving ultrasound-mediated BBB disruption in animal and clinical studies, as well as the efficacy and safety methods and outcomes associated with each. A systematic search of electronic databases helped in identifying relevant, included studies. Reference lists of included studies were further screened to identify supplemental studies for inclusion. In total, 107 articles were included in this review, and the following parameters were identified as influencing efficacy and safety outcomes: microbubbles, transducer frequency, peak-negative pressure, pulse characteristics, and the dosing of ultrasound applications. Current protocols and parameters achieving ultrasound-mediated BBB disruption, as well as their associated efficacy and safety outcomes, are identified and summarised. Greater standardisation of protocols and parameters in future preclinical and clinical studies is required to inform robust clinical translation.

## 1. Introduction

### 1.1. The Blood–Brain Barrier and Drug Delivery

The blood–brain barrier (BBB) is a selectively permeable structure that restricts the passage of solutes from the brain’s microvasculature into its extracellular space. Anatomically, the BBB is composed of the apical and basal membranes of the cerebrovascular endothelial cells (CECs), an associated basement membrane containing embedded pericytes, and perivascular foot-like processes of astrocytes ensheathing the abluminal capillary surface collectively referred to as the neurovascular unit [1] (see Figure 1). The CECs express a limited range of membrane carrier proteins and numerous membrane efflux pumps and have a dense number of tight junctions linking them to other CECs. This combined structural arrangement restricts the movement of most hydrophilic and high molecular weight molecules (exceeding 400 to 500 Da) [2], inflammatory cells, and pathogens, thereby providing a vital function in maintaining homeostasis and preventing the entry of harmful substances into the brain. The BBB, therefore, grants a significant survival advantage, but it also poses a disadvantage in its inability to allow most therapeutic agents to penetrate it, rendering 98% of small-molecule agents and 100% of large-molecule agents unable to enter the brain parenchyma [3]. As a result, there is a significant limitation to the pharmacological agents available in the treatment of central nervous system (CNS) conditions, including brain malignancies, dementias, and other neurodegenerative conditions.

### 1.2. Ultrasound-Mediated Drug Delivery and the BBB

An approach to overcoming the challenge posed by the BBB is to temporarily induce BBB disruption, in a controlled and targeted manner, to enhance the uptake of therapeutic agents into desired target locations in the brain. A minimally invasive strategy that has been employed to achieve this is via the application of ultrasound. The use of transcranial ultrasound for enhancing drug delivery in the CNS also extends beyond BBB disruption, including in functioning as an external trigger to initiate drug release from nanoparticles [4] at targeted areas of the BBB in diseases such as epilepsy [5]. While the use of ultrasound in disrupting the BBB was first described in the 1950s [6], it is within the last 20 years [7] that this technique has garnered significant research interest in improving drug delivery to the brain. Subsequently, numerous preclinical animal studies, as well as several phase I/II clinical trials [8,9,10,11] (ClinicalTrials.gov identifier numbers: NCT03321487, NCT03322813, NCT02253212, NCT03608553, NCT03626896, NCT02343991, NCT02986932, NCT03712293), assessing the feasibility of this drug delivery approach have emerged. While the exact mechanisms underlying ultrasound-mediated BBB disruption are not yet well defined, the mechanical bioeffects of ultrasound exposure are thought to predominate. When exposed to ultrasound, dissolved gas bubbles within the vasculature experience a phenomenon known as cavitation, where they experience oscillatory changes in their volume, expanding in volume with rarefactions (low sonic pressure), and contracting with compressions (high sonic pressure) of the ultrasound waves [7]. Gas bubbles may also experience an acoustic radiation force, where they gain additional translational movement towards the direction of the ultrasound beam. These effects together are thought to contribute to the observed reduction in tight junction proteins between CECs of the BBB [12], the reduced expression of P-glycoprotein drug efflux pumps along the CEC membrane [13,14,15], and the increased formation of transcytotic vesicles across the CECs [16]. To enhance these mechanical effects ultrasound contrast agents or microbubbles (preformed gas-filled bubbles, typically of 1 to 6 µm in diameter) are co-administered, thereby increasing the number of available echogenic centres that can induce a mechanical effect within the cerebral vasculature, and thus reducing the threshold of ultrasound intensity required for BBB permeabilisation. Furthermore, magnetic resonance imaging (MRI) has been coupled with ultrasound and microbubbles, as a way of ensuring precise targeting [17] of the ultrasound beam to specific regions within the brain, confining drug penetration to these foci only, e.g., tumour sites for glioma patients.

### 1.3. Challenges with Ultrasound-Mediated BBB Disruption

Though numerous studies have achieved variable levels of successful ultrasound-mediated BBB disruption, it has become clear that the extent of BBB disruption is greatly influenced by the choice of ultrasound parameters, as well as the type and dose of microbubbles used alongside each sonication [18]. Unfortunately, ultrasound-mediated BBB disruption has potential adverse effects, including haemorrhagic change (ranging from sparse erythrocyte extravasation to gross intracerebral haemorrhaging) [19], oedema [9,20,21,22,23,24,25,26], inflammation [26,27], neuronal ischaemia [28,29], and tissue apoptosis [23,27,28,30,31,32,33,34]. The occurrence of these adverse effects is theorised to be due to excessive mechanical activity, where sonicated bubbles within the vasculature experience inertial, non-stable cavitation, rapidly imploding to exert excessive endothelial force, causing the extravasation of fluid, erythrocytes, and leucocytes into the surrounding tissue. Additionally, these effects may contribute to direct neuronal and glial injury. This has precipitated a large body of both preclinical and clinical studies demonstrating ultrasound-mediated BBB disruption with a range of sonication parameters, accompanied by a diverse set of reported safety outcomes. Ultimately, this has made selecting an appropriate sonication protocol that provides both successful and safe BBB disruption a difficult task. Previous narrative reviews of this body of evidence have been conducted but only broadly summarise key ultrasound-related parameters and their associated effects on the efficacy and safety of ultrasound-mediated BBB disruption. A systematic review of such literature and a published database of individual sonication paradigms and their consequential safety outcomes has yet to be conducted. The results of this systematic review will aim to inform future researchers of methods, sonication protocols, and the parameters that influence BBB disruption, as well as the safety outcomes associated with each, across a variety of experimental models (rodents, rabbits, sheep, pigs, non-human primates (NHPs)) and humans.

## 2. Materials and Methods

The systematic review was conducted according to the Preferred Reporting Items for Systematic Reviews and Meta-analyses (PRISMA) guidelines [35,36], PRISMA’s checklist or PRISMA Flow Diagram for systematic reviews has been completed and is available as Appendix A in Appendix A. This systematic review was not registered.

### 2.1. Eligibility Criteria

All studies with (i) clearly outlined sonication protocols (containing the relevant terms below), where (ii) ultrasound was applied to the brain of in vivo animal or human participants and where (iii) successful BBB disruption was achieved and confirmed using a reliable method, were included within this systematic review. Additionally, included studies had to have conducted appropriate safety assessments and reported any adverse effects associated with each protocol achieving successful BBB disruption. Only sonication protocols achieving confirmed BBB disruption, with corresponding safety assessments, were included within the data collection process. Protocols where the primary aim was applying ultrasound for cellular, viral, or gene delivery; neuromodulation; stimulation; or tissue ablation were excluded. Studies not published in English and review papers were also excluded.

### 2.2. Information Sources

This systematic review was based on searches from the following online search databases: PubMed, Medline, and EMBASE. A complete search of databases was conducted on 22 September 2021 to include any subsequent articles published within this timeframe. Additional articles were identified from reviewing the reference lists of relevant articles identified via the initial database search, as well as via Google Scholar alerts from the date of the initial search. A complete search of databases was conducted by K.G.

### 2.3. Search Strategy

The following MeSH terms were used to identify relevant articles:‘Ultrasound’ OR ‘focused ultrasound’ OR ‘MRI-guided ultrasound’ OR ‘MR-guided focused ultrasound’;‘Blood brain barrier’ OR ‘BBB’;‘Disruption’ OR ‘permeabilisation’ OR ‘permeabilization’ OR ‘opening’;‘Drug delivery’.

### 2.4. Study Selection

Citations identified from the searches of the three databases were collated into a combined EndNote (X9) library and were de-duplicated as per a published protocol [37]. All unique citations were then screened for inclusion according to the eligibility criteria. Citation screening was conducted by K.G. in the following sequence: initially by their titles, then by their abstracts, and finally by reviewing the entire text. K.G., A.B.-F., and A.B. independently identified additional studies from Google Scholar alerts and from reviewing the reference list of relevant articles identified via the database search.

### 2.5. Data Items and Collection Process

From each included study, the following data items were extracted: (1) species of in vivo subject; (2) type of ultrasound transducer used; (3) methods for assessing BBB disruption; (4) methods for assessing safety; (5) microbubbles used (type, dose, administration protocol); (6) sonication parameters (frequency, peak negative pressure in situ, continuous or pulsated delivery, sonication duration, number of sonications, interstimulus interval, number of independent sessions, intersession interval). The extracted data items were then tabulated (Appendix A—Appendix A). Additional summary tables regarding methods used to assess BBB disruption efficacy Table 1) and safety, as well as microbubble and ultrasound parameters influencing ultrasound-mediated BBB disruption are also included in this systematic review. K.G. conducted all data extraction and collection.

## 3. Results

### 3.1. Included Studies

A total of 1480 citations were identified after the initial literature search from the three databases (Figure 2). After de-duplication, 882 unique citations were identified and were subsequently screened, in order, by their title and abstract, and then by a full-text review of the remaining articles (*n* = 100). After completing the screening process, a total of 76 studies were identified and included for data extraction and synthesis. An additional 31 studies were identified and included from Google Scholar alerts and after screening reference lists of included studies. Ultimately, a total of 107 studies were eligible for inclusion in our qualitative comparison and analysis.

### 3.2. Ultrasound Devices

A range of commercial (e.g., FUS Instruments, Imasonic, Riverside Research Institute, Sonic Concepts) and in-house manufactured ultrasound devices were utilised in the included study protocols, with the majority of these being single-element or single piezoelectric devices. In recent times, multi-element devices have been developed to overcome associated concerns around ultrasound attenuation and beam defocussing, providing better transcranial transmission. These devices have been tested in preclinical animal models as well as in many ongoing and published clinical trials. In addition, we identified a protocol utilising two single-element transducers in tandem [38], and two that even used diagnostic [39,40], imaging transducers to disrupt the BBB. A comparative figure highlighting therapeutic ultrasound devices identified amongst included literature is shown in Figure 3.

#### 3.2.1. Single-Element Ultrasound Devices

While single-element devices are smaller, and more accessible for testing in preclinical studies, their application in clinical studies has been more limited due to the attenuation and defocussing resulting from the application of a single ultrasound beam [41]. A strategy employed [9] to circumvent this issue has been to create a bony window in the calvaria and apply ultrasound directly through it, thereby reducing attenuation of the ultrasound beam at the bone interface [42]. Two animal studies directly compared this strategy with application through intact bone, with one highlighting no difference with the use of a 260 kHz transducer in rabbits [30] and the other highlighting significant improvements with the use of a 28 kHz transducer in pigs [23]. The SonoCloud-1^®^, a single-element device manufactured by CarThera, is one such example that requires implantation via a burr hole in the skull. Currently, this is the only implantable device we identified amongst all included protocols, and it has demonstrated efficacy in disrupting the BBB in both animal [20,43,44] and clinical studies [9]. Dual single-element transducers were employed in a couple of studies [38,45] to initiate BBB disruption using a lower frequency transducer, while a higher frequency transducer was employed in an attempt to stimulate the transport of a therapeutic agent into the brain parenchyma.

#### 3.2.2. Multi-Element Ultrasound Devices

Multi-element array devices confer the benefit of being able to treat multiple, separate tissue foci simultaneously, as well as providing greater spatial coverage than single-element devices [46]. Additionally, multi-element phased array devices confer the ability to alter the phase and amplitude of individual transducers, correcting for aberrations to the ultrasound beam as it surpasses more complex skull surfaces, such as the human calvaria [46,47]. An emerging leader in this category of devices is the ExAblate^®^ system (manufactured by InSightec), a 1024-element phased array device, initially designed for the thermal ablation of tissue, prior to being applied in rat [48,49], NHP [50], and clinical studies [8,10,11,51,52,53] to disrupt and open the BBB. The ExAblate^®^ system functions with a large, stereotactically positioned helmet that is coupled to an MRI system to help plan sonication targets and monitor the procedure. A second multi-element phased array system—NaviFUS^®^ (produced by NaviFUS corporation)—is a 256-element phased array device that has also demonstrated its ability to disrupt the BBB in a recent clinical trial [54]. While this device and the ExAblate^®^ share similarities in their appearance, NaviFUS^®^ does not require stereotactic positioning and relies on traditional, neurosurgical, navigation to help plan and guide ultrasound beams to sonication targets. The SonoCloud-9^®^ (manufactured by CarThera) is an iteration of the SonoCloud-1^®^ and functions as an implantable grid of nine interconnected transducer elements. As yet, no published study has employed the SonoCloud-9^®^ for in vivo BBB disruption, but ongoing clinical trials exist (NCT03744026, NCT04614493, NCT04528680).

#### 3.2.3. Diagnostic Ultrasound Devices

Disruption of the BBB was also achieved using diagnostic ultrasound imaging devices in mice [39,40], albeit with much broader and less well-defined tissue coverage as compared to the prior mentioned therapeutic ultrasound devices. This is partly due to the higher central frequencies (2–10 MHz) that diagnostic devices tend to operate with, often resulting in greater aberration and defocussing of the penetrating ultrasound beam [39].

#### 3.2.4. Implantable Ultrasound Devices

Nearly all identified ultrasound devices were non-implantable and usually required precise (e.g., stereotactic, neuronavigation, MRI-guided) positioning over the cranium of each subject prior to sonication. Implantable devices (SonoCloud^®^ devices) are highly portable and eliminate the need for repeat repositioning, at the cost of requiring more invasive, surgical placement. Non-implantable devices have the advantage of being surgically non-invasive and can be repositioned for targeting multiple sites, at the cost of longer ultrasound sessions, during which subjects are not ambulatory [55]. A recent review [55] suggested that non-implantable, extracranial devices were not appropriate for targeting superficial lesions, a claim not supported by findings from identified protocols that achieved successful and safe BBB disruption in superficial cortical regions (e.g., primary motor cortex [8,56], primary visual cortex [24], prefrontal cortex [10,57]) using both single-element and multi-element phased array devices.

### 3.3. Methods for Assessing Successful BBB Disruption and Opening

#### 3.3.1. MRI

The majority of identified protocols confirmed BBB disruption and its subsequent opening via contrast-enhanced T1-weighted MRI (CE-T1 MRI). The fundamentals of CE-T1 MRI involve taking T1 images prior to sonication, administering a gadolinium contrast agent, and then acquiring T1 images post-sonication. The molecular weight (~0.5 to 1.1 kDA) and hydrophilicity of gadolinium contrast agents make them incapable of passing the BBB in normal circumstances. Therefore, an opening in the BBB will result in visible extravasation of these agents into the cerebral interstitium, marked by hyperintensity on a post-sonication T1 image (see Figure 4). MRI quantification methods used in identified studies can be broadly summarised as follows: (1) T1 mapping to estimate the concentration and spatial distribution of the contrast agent [58,59]; (2) calculating vascular transfer coefficients of contrast agents after dynamic contrast-enhanced T1 imaging [60]; (3) or calculating changes in contrast signal enhancement [31]. The ability to observe BBB disruption in vivo is conferred by CE-T1 MRI, without requiring postmortem histological analysis. Multiple investigations have shown correlative relationships between the extravasation of the MRI contrast agent and that of histological tracers [61,62] and some therapeutic agents, including Herceptin [31], doxorubicin [63], and nanoparticles [23,38]. A few studies have employed the use of T2/T2* weighted MRI to track the uptake of superparamagnetic iron oxide (SPIO)-labelled drug molecules into sonicated brain parenchyma. This MRI technique provides a more sensitive method for assessing drug uptake by allowing the real-time, direct visualisation [23,64,65] of drug extravasation into the brain, as opposed to the use of a proxy marker (gadolinium contrast agent). Additionally, studies have used CE-T1 MRI to track the reversibility of BBB opening after sonication, unsurprisingly concluding that the duration of BBB opening, and thus the reversal time, increases with the initial degree of BBB disruption [66].

#### 3.3.2. Tracer Molecules

Tracer molecules, including Evans/trypan blue (67 kDa when albumin-bound), fluorescein (333 Da), fluorescently labelled dextrans (3 to 2000 kDa), horseradish peroxidase (44 kDa), and antibodies (either endogenous or exogenously administered), are molecules incapable of surpassing the BBB and are widely administered to assess its opening. These molecules are readily available and cheap and can be observed macroscopically (Evans and trypan blue) or microscopically (fluorescein, dextrans, horseradish peroxidase), and their cerebral uptake can be quantified to confirm the degree of successful BBB opening. Additionally, tracer molecules come in various molecular weights, meaning their extravasation can better delineate the size of molecular weight therapeutics that could pass the disrupted BBB. Rodent studies [67,68] have highlighted differences in the extravasation of variable-sized dextrans after BBB opening with equivalent parameter sonications, where lower molecular weight dextrans (3 to 70 kDa) have significant extravasation, while higher (500 to 2000 kDa) weight dextrans have minimal extravasation. Therefore, the use of variable molecular weight tracer molecules gives an advantage over CE-T1 MRI, the latter of which only indicates BBB opening to a potential maximum threshold equal to the molecular weight of the injected gadolinium contrast agent. Due to the tissue analysis required for assessing tracer uptake, these methods are almost exclusively used in preclinical, animal studies and are harder to conduct in human trials due to the necessity of a brain biopsy. Interestingly, one clinical trial [51] did microscopically assess fluorescein uptake into resected, sonicated tumour/peritumoral tissue, reporting a 2.2-fold increase in comparison to non-sonicated tumour tissue.

#### 3.3.3. Therapeutic Agent Quantification

Another approach in assessing BBB opening is to directly assess the cerebral uptake of normally impenetrable therapeutic agents, e.g., antibodies and chemotherapeutic agents. This has been done by quantifying the concentration of therapeutics from tissue homogenates via high powered liquid chromatography, liquid chromatography–mass spectrometry, or fluorometry or by labelling therapeutic molecules with fluorescent [38,69], radioactive, or magnetic markers [23] in order to more sensitively visualise the extent of tissue penetration and the location of therapeutic accumulation. Ultimately, this latter approach is the most direct method of ascertaining the clinical efficacy of ultrasound as a novel technique for enhancing therapeutic delivery to the brain. Only one [11] identified clinical trial reported any data on quantified therapeutic uptake after BBB disruption. More of these investigations are required to supplement concurrent CE-T1 MRI assessments in human trials to further validate the efficacy of this novel approach to drug delivery. Furthermore, the timing of drug administration relative to the application of ultrasound appears to influence drug uptake into the targeted brain region [70,71].

#### 3.3.4. Comparing BBB Disruption between Studies

While a diverse range of reliably proven methods for assessing BBB opening exist, a standard protocol for conducting each does not, complicating the comparative analysis of successful BBB opening between studies. Of note amongst included studies was the variation in dose, administration time, and route of delivery of contrast agents, tracer dyes, exogenous antibodies, and/or therapeutic agents between studies. Investigations have noted significant variation in the extravasation of these agents as a response to altered administration times [37,66] relative to each sonication and to the route of chosen delivery [72] (e.g., intravenous vs. intraperitoneal). In addition, there exists a range of MRI parameters and quantification methods for the uptake of contrast and tracer agents utilised between studies, further complicating the ability to perform external comparisons between study protocols. As a result of the observed heterogeneity in specific BBB disruption assessment protocols, the ‘Comparative Degree of Observed BBB Disruption’ column (Appendix A—Appendix A) qualitatively highlights relative differences in BBB disruption achieved between different parameters investigated within the same study, as opposed to parameters between different studies. A comparison of methods used to assess BBB disruption across included protocols is presented in Table 1.

**Table 1 pharmaceutics-14-00833-t001:** Summary of the methods used by included protocols in assessing the extent of BBB disruption.

In Vivo Subject	Study and Year Published	Assessments of BBB Disruption and Opening							
		MRI	Tracer Molecules						Quantified Therapeutic Uptake
			EB	TB	Fl	FD	HRP	Antibodies	
Mouse	Baghirov et al., 2018 [38]	**X**							**X** (Polymeric nanoparticles)
	Baseri et al., 2010 [73]	**X**				**X**			
	Bing et al., 2009 [39]	**X**							
	Chen et al., 2013 [74]					**X**			
	Chen et al., 2014 [67]					**X**			
	Choi et al., 2010 [75]					**X**			
	Choi et al., 2011 [76]					**X**			
	Choi et al., 2011 [60]	**X**				**X**			
	Choi et al., 2008 [77]	**X**							
	Choi et al., 2010 [68]					**X**			
	Englander et al., 2021 [78]	**X**	**X**						**X** (Etoposide)
	Jordao et al., 2013 [61]	**X**						**X** (Anti-endogenous IgG and IgM)	
	Kinoshita et al., 2006 [31]	**X**		**X**					**X** (Herceptin)
	Kinoshita et al., 2006 [62]	**X**		**X**				**X** (Anti-D4 IgG)	
	Lapin et al., 2020 [79]	**X**							
	Liu et al., 2014 [80]	**X**	**X**						**X** (Temozolomide)
	McDannold et al., 2017 [81]	**X**							
	McMahon et al., 2020 [59]	**X**				**X**		**X** (Anti-albumin IgG)	
	Morse et al., 2022 [82]								**X** (Fluorescently labelled, unloaded liposomes)
	Morse et al., 2019 [83]					**X**		**X** (Anti-albumin IgG)	
	Olumolade et al., 2016 [84]	**X**							
	Omata et al., 2019 [85]		**X**			**X**			
	Raymond et al., 2007 [86]		**X**			**X**			
	Raymond et al., 2008 [87]	**X**	**X**	**X**				**X** (Anti-amyloid + anti-endogenous IgG)	
	Samiotaki et al., 2012 [66]	**X**							
	Shen et al., 2016 [69]		**X**						**X** (Fluorescently labelled, unloaded liposomes)
	Sierra et al., 2017 [88]	**X**			**X**				
	Vlachos et al., 2011 [72]	**X**							
	Wu et al., 2014 [89]								**X** (Liposomal doxorubicin)
	Zhang, D. et al., 2020 [43]			**X**					**X** (Paclitaxel—free and protein-bound)
	Zhao, B. et al., 2018 [40]		**X**						
Rat	Ali et al., 2018 [90]	**X**	**X**						**X** (Doxorubicin)
	Aryal et al., 2017 [15]	**X**		**X**					
	Aryal et al., 2015 [91]	**X**		**X**					**X** (Doxorubicin)
	Aryal et al., 2015 [70]	**X**		**X**					**X** (Doxorubicin)
	Aslund et al., 2017 [92]	**X**							**X** (Pegylated macromolecule)
	Cho et al., 2016 [14]	**X**	**X**						
	Chopra et al., 2010 [93]	**X**							
	Fan et al., 2016 [64]	**X**							**X** (SPIO-labelled, doxorubicin-loaded microbubbles)
	Fan et al., 2014 [45]		**X**						**X** (Carmustine loaded microbubbles)
	Fan et al., 2015 [94]		**X**						**X** (Carmustine loaded microbubbles)
	Goutal et al., 2018 [95]	**X**	**X**						
	Han et al., 2021 [96]	**X**							
	Huh et al., 2020 [97]	**X**							
	Jung et al., 2019 [98]	**X**	**X**						**X** (Doxorubicin)
	Kobus et al., 2016 [99]	**X**							
	Kovacs et al., 2017 [27]	**X**						**X** (Anti-albumin IgG)	
	Kovacs et al., 2018 [100]	**X**							
	Liu et al., 2009 [65]	**X**	**X**						
	Liu et al., 2010 [101]	**X**	**X**						**X** (Carmustine)
	Liu et al., 2010 [102]		**X**						
	Liu et al., 2008 [32]	**X**	**X**						
	Liu et al., 2010 [103]	**X**							
	Marty et al., 2012 [58]	**X**							
	McDannold et al., 2019 [48]	**X**							**X** (Carboplatin)
	McDannold et al., 2020 [49]	**X**							**X** (Irinotecan and SN-38)
	McDannold et al., 2011 [104]	**X**		**X**					
	Mcmahon et al., 2017 [26]	**X**							
	Mcmahon et al., 2020 [105]	**X**	**X**						
	Mcmahon et al., 2020 [106]	**X**							
	O’Reilly et al., 2017 [107]	**X**							
	O’Reilly et al., 2011 [108]	**X**							
	Park et al., 2017 [109]	**X**		**X**					**X** (Doxorubicin)
	Park et al., 2012 [71]	**X**		**X**					**X** (Doxorubicin)
	Shin et al., 2018 [19]		**X**						
	Song et al., 2017 [110]		**X**						
	Treat et al., 2007 [63]	**X**	**X**						**X** (Doxorubicin)
	Tsai et al., 2018 [33]		**X**						
	Wei et al., 2013 [111]	**X**	**X**						**X** (Temozolomide)
	Wu et al., 2017 [112]	**X**	**X**						
	Yang et al., 2013 [113]	**X**	**X**						
	Yang et al., 2014 [114]	**X**	**X**						
	Yang et al., 2012 [34]	**X**	**X**						
	Yang et al., 2011 [115]	**X**	**X**						
	Yang et al., 2012 [116]	**X**	**X**						
	Zhang, Y. et al., 2016 [117]	**X**							
Rabbit	Beccaria et al., 2013 [20]	**X**	**X**						
	Chopra et al., 2010 [93]	**X**							
	Hynyen et al., 2005 [28]	**X**					**X**		
	Hynyen et al., 2006 [30]	**X**					**X**		
	McDannold et al., 2006 [118]	**X**							
	McDannold et al., 2007 [25]	**X**							
	McDannold et al., 2008 [119]	**X**							
	McDannold et al., 2008 [120]	**X**							
	Mei et al., 2009 [121]	**X**	**X**						**X** (Methotrexate)
	Wang et al., 2009 [122]	**X**	**X**						
Dog	O’Reilly et al., 2017 [123]	**X**							
Pig	Liu et al., 2011 [23]	**X**	**X**						**X** (SPIO nanoparticles)
Sheep	Pelekanos et al., 2018 [29]		**X**					**X** (Anti-endogenous IgG)	
	Yoon et al., 2019 [124]	**X**							
Non-Human Primate (NHP)	Arvantis et al., 2012 [50]	**X**							
	Downs et al., 2015 [21,22]	**X**							
	Goldwirt et al., 2016 [44]	**X**							**X** (Carboplatin)
	Horodyckid et al., 2017 [56]	**X**							
	Marquet et al., 2014 [125]	**X**							
	Marquet et al., 2011 [24]	**X**							
	McDannold et al., 2012 [126]	**X**		**X**					
	Pouliopoulos et al., 2019 [57]	**X**							
	Wu et al., 2016 [127]	**X**							
Human	Abrahao et al., 2019 [8]	**X**							
	Anastasiadis et al., 2021 [51]	**X**			**X**				
	Chen et al., 2021 [54]	**X**							
	Gasca-Salas et al., 2021 [52]	**X**							
	Idbaidh et al., 2019 [9]	**X**							
	Lipsman et al., 2018 [10]	**X**							
	Mainprize et al., 2019 [11]	**X**							**X** (Liposomal doxorubicin and temozolomide)
	Park et al., 2020 [53]	**X**							

***EB:***
*Evans blue; **TB:** trypan blue; **FL:** fluorescein; **FD:** fluorescently labelled dextrans; **HRP:** horseradish peroxidase.*

### 3.4. Methods of Assessing Safety Outcomes

The safety of ultrasound-mediated BBB disruption is crucial for this technology to receive mainstream clinical adoption in the treatment of CNS disease; thus, in this review, it was essential to only include ultrasound protocols with a corresponding safety assessment. The techniques employed by studies to analyse safety outcomes can be broadly characterised into five categories: macroscopic, histological, biochemical, electrophysiological, and behavioural safety assessments. Histological and macroscopic assessments have undoubtedly been the most extensively conducted techniques amongst included literature, as they highlight detailed changes in tissue architecture and can be readily conducted in preclinical animal studies. When comparing safety outcomes between studies employing different ultrasound protocols and parameters, the time of safety data acquisition is vital [17]. Studies have highlighted how MRI and histological adverse safety events may progress or regress with the time interval from sonication to MRI acquisition [32,71] or tissue extraction [59,88]. For this reason, we have included, when available, the timing of MRI or histological safety data acquisition from the last sonication, for each included protocol within this study (Appendix A). A detailed summary of specific safety investigations employed across included studies is presented in Table 2.

#### 3.4.1. Macroscopic Assessments

MRI techniques, most commonly T2, T2*, and susceptibility-weighted imaging (SWI), have been employed to detect evidence of oedematous (hyperintensities on T2) [21,22] and haemorrhagic (hypointensities on T2* and SWI) [50] change within the sonicated brain. Currently, the clinical application of ultrasound-mediated BBB disruption has relied on MRI, serving as an assessment technique for confirming in vivo BBB opening (as previously discussed). It also provides the ability to observe changes in tissue health and to track the progression of any of these changes serially, for hours and days following sonication [10,21,22,80]. In clinical trials, ultrasound-mediated BBB disruption has been generally well tolerated, but MRI findings have also shown transient oedematous [8,9] and microhaemorrhagic change [10], observed in a small subset of patients only. A few rodent studies [27,103] also utilised T2* MRI to image the extravasation of superparamagnetic-labelled macrophages into the sonicated tissue when assessing for an inflammatory reaction to ultrasound-mediated BBB disruption. Thermometry is another macroscopic safety assessment identified amongst protocols, playing a role in the monitoring of unwanted thermogenic bioeffects from ultrasound application. Methods of thermometry included ex vivo calvaria thermometry [29], the use of in situ thermal probes [89,98], and real-time MR thermometry [8,10,52]. Generally, temperature elevations did not exceed 1.5 °C in most studies employing in vivo thermometry [8,10,52,98]. One study investigated the application of continuous ultrasound to induce a hyperthermic effect in mice, noting a temperature elevation of 13 °C over a 10 min period of sonication [89]. Positron emission tomography (PET) scanning has also been conducted in a handful of NHP and clinical studies, revealing no changes in glucose uptake and metabolism [52,56] in sonicated tissue following multiple ultrasound sessions. Direct visualisation (without imaging) of gross haemorrhage in brain tissue has been reported with significant BBB disruption in animal studies [19] after applying more intense (higher pressure) ultrasound or over prolonged sonication periods.

#### 3.4.2. Histological Assessments

Basic histological stains, most notably haematoxylin and eosin, Cresyl violet/Luxol fast blue, and Perl’s Prussian Blue, have been readily utilised to confirm microscopic changes to tissue architecture, haemorrhagic change, and iron deposition. Immunolabelling of specific proteins has allowed for the investigation of more specific histological changes associated with ultrasound-mediated BBB disruption, including potential reactive astrogliosis (glial fibrillary acid protein or GFAP), microglial activation (Iba-1), and neurogenesis (BrdU) macrophage (CD68+) and T lymphocyte (ICAM-1, CD-4, CD-8) infiltration. Additionally, immunolabelling of endothelial markers (RECA-1 and CD-31) has allowed for the screening of direct endothelial damage [14] and assessing tissue vascular density when comparing BBB opening between sonicated tumour and normal tissue [54]. Major reported histopathological findings include a continuum of haemorrhagic change within the brain parenchyma [19], oedema [9,20,21,22,23,24,25,26], neuronal ischaemia [28,29], tissue apoptosis [23,27,28,30,31,32,33,34], immune cell infiltration, and gliosis. Reported histopathological outcomes have been identified at a variety of endpoints, from immediately following [115] to months after initial sonication [123], highlighting both the potential acuteness and chronicity at which ultrasound-mediated BBB disruption may exert unwanted biological effects. Histopathological assessments generally reinforce pathological findings on MRI, but in some studies [62,92,93], they appear to highlight pathological change in the absence of any on MRI, despite equivalent timing of data acquisition, suggesting higher sensitivity for adverse pathological change.

#### 3.4.3. Biochemical Assessments

Biochemical assessments, namely polymerase chain reaction, Western blotting and enzyme-linked immunosorbent assays, have been employed in a handful of rat investigations [26,27,100,105] to track changes in the expression of proinflammatory genes and proteins following ultrasound-mediated BBB disruption. Of note, these studies have shown an upregulation in the transcriptomic expression of proinflammatory genes related to the NF-kB [26,27] (e.g., Ccl2, Ilα, Ilβ, Selp, Tnf, Icam1) and AkT/GSK*β* pathways [27] with larger doses of administered microbubbles. Furthermore, temporal changes in the proteomic expression of Iba1 (activated microglia) and GFAP (reactive astrocyte marker) have been described over a time course of 15 days [61]. Serum biochemical analysis was employed as a safety assessment in one study, which reported an increase in fibrinogen levels 8 days after sonication in animals exposed to the highest intensity ultrasound, likely attributable to the corresponding histological findings [33]. These findings have generated a link between ultrasound-mediated BBB disruption and subsequent proinflammatory changes, and further work is required in assessing the significance and potentially deleterious effect this may have on the health of the sonicated brain.

#### 3.4.4. Electrophysiological Assessments

Electrophysiological investigations following ultrasound-mediated BBB disruption, including electroencephalography, electromyography, and somatosensory evoked potentials, have been occasionally used within the identified literature, appearing in two NHP [21,56] studies and a clinical trial [8]. One NHP study reported no abnormal electroencephalographic waveform changes, nor any to the somatosensory evoked potentials from the median or popliteal nerves, following repeated BBB disruption of the primary motor cortex over a 15-day period [56]. No differences in electromyographic signals from the temporalis muscle of NHPs following repeat BBB disruption of basal ganglia structures were noted in another investigation [21] either. In a clinical trial involving BBB disruption of the primary motor cortex in ALS patients, electroencephalographic readings also remained unchanged after repeat sonications [8]. Although reassuring, these data are generated from a very small sample size. Given the emerging role of ultrasound in the field of neuromodulation [128], it is surprising to see the relative lack of neuro-electrophysiological analyses conducted across current literature.

#### 3.4.5. Physical and Behavioural Assessments

Physical and behavioural assessments have been employed to monitor safety outcomes following ultrasound-mediated BBB disruption in a range of experimental models, including in rodent, dog, NHP, and human studies. Adverse motor outcomes in rodents have been assessed via rotarod and pinch grip tests, as well as via gross motor observations, and outcomes have included periods of hypoactivity, tremor, and ataxia in rodents [33,84] that underwent higher intensity sonications. Conversely, glioma-implanted rodents sonicated with similar intensity ultrasound and lower microbubble doses have exhibited no changes in motor coordination or grip strength following ultrasound with lower intensity ultrasound [78,90]. Other reported motor outcomes include mildly altered reaction times in one NHP study [21] and reversible, mild upper-limb hemiplegia in another NHP study [24]. Physiological outcomes have been reported, including transient, microbubble-associated tachycardia [78] and tachypnoea [56] in some rat and NHP studies, but these do not appear to be corroborated by other preclinical [22] and clinical studies. Detailed neurological testing following sonication in aged canines has also been conducted, yielding no changes in neurological or mental status [123]. Additionally, long-term cognitive testing in NHPs [22] via reward-based reaction and visual dot motion tasks has been conducted, revealing no significant changes to cognitive decision-making abilities, but potentially eliciting a reduction in motivation. Overall, motor and behavioural changes following BBB disruption in preclinical models appear to be mild.

In clinical trials, physical findings most frequently included pain associated with setting up and stabilising the patient’s head into the phased array transducer [8,10,11] or minimal irritation from connecting the implanted transducer to its electrical supply [9]. In one trial [9], a single patient experienced a transient facial palsy that occurred immediately following three separate sonications, resolving within two hours after steroid administration. Clinical trials have also incorporated neuropsychological assessments (e.g., Mini Mental State Exam, Montreal Cognitive Assessment) to assess potential alterations in cognition after BBB disruption in patients with Parkinson’s disease dementia [52], Alzheimer’s dementia [10], and ALS [8]. In summary, the occurrence of adverse physical and behavioural outcomes following ultrasound-mediated BBB disruption in humans has been infrequent and predominantly transient when present. Once again, these data are limited due to small patient sample sizes and the lack of sham or control groups. Additionally, significant patient neurological comorbidity in these trials makes it difficult to directly attribute adverse events to ultrasound-mediated BBB disruption.

**Table 2 pharmaceutics-14-00833-t002:** Summary of safety assessments conducted by included protocols.

In Vivo Subject	Study Reference	Safety Assessments															
		Macroscopic				Histological									Biochemical	Electrophysiological	Physical/Behvaioural
		MRI	PET	Δ*T*	Gross	H/E	TUNEL	VF	LB	CV	PB	GFAP	Iba1	Other			
Mouse	[38]	**X**				**X**											
	[73]	**X**			**X**	**X**											
	[39]					**X**											
	[74]					**X**											
	[67]					**X**											
	[75]					**X**											
	[76]					**X**											
	[60]					**X**	**X**										
	[77]	**X**				**X**											
	[68]					**X**											
	[78]	**X**				**X**											**X**
	[61]											**X**	**X**	**X**	X (PCR + WB)		
	[31]				**X**	**X**	**X**	**X**									
	[62]				**X**	**X**											
	[79]	**X**															
	[80]					**X**											
	[81]					**X**											
	[59]	**X**				**X**											
	[82]					**X**											
	[83]					**X**											
	[84]					**X**			**X**	**X**							**X**
	[85]					**X**								**X**			
	[86]				**X**	**X**								**X**			
	[87]					**X**											
	[66]					**X**											
	[69]					**X**											
	[88]	**X**				**X**							**X**				
	[72]	**X**				**X**											
	[89]			**X**	**X**		**X**										
	[43]					**X**											
	[40]					**X**								**X**			
Rat	[90]					**X**								**X**			**X**
	[15]	**X**				**X**											
	[91]	**X**				**X**											
	[70]	**X**				**X**											
	[92]	**X**				**X**											
	[14]	**X**				**X**								**X**			
	[93]	**X**				**X**											
	[64]					**X**											
	[45]	**X**				**X**	**X**										
	[94]					**X**											
	[95]	**X**	**X**														
	[96]	**X**				**X**						**X**		**X**	X (AQP-4)		
	[97]					**X**											
	[98]	**X**		**X**		**X**											
	[99]	**X**				**X**						**X**					
	[27]					**X**	**X**					**X**	**X**	**X**	X (ELISA, PCR, WB)		
	[100]	**X**				**X**						**X**	**X**	**X**	X (WB)		
	[65]	**X**			**X**	**X**					**X**						
	[101]	**X**				**X**											
	[102]					**X**											
	[32]	**X**				**X**	**X**										
	[103]	**X**									**X**			**X**			
	[58]	**X**															
	[48]	**X**				**X**			**X**								
	[49]	**X**				**X**											
	[104]				**X**	**X**											
	[26]	**X**				**X**									X (PCR)		
	[105]	**X**				**X**									X (PCR)		
	[106]	**X**										**X**		**X**	X (ELISA)		
	[107]	**X**				**X**											
	[108]	**X**				**X**											
	[109]	**X**				**X**											
	[71]	**X**				**X**											
	[19]				**X**	**X**											
	[110]				**X**									**X**			
	[63]				**X**	**X**											
	[33]					**X**	**X**					**X**			X (Plasma fibrinogen)		**X**
	[111]	**X**				**X**											
	[112]	**X**				**X**											
	[113]					**X**											
	[114]					**X**											
	[34]					**X**	**X**										
	[115]					**X**											
	[116]					**X**	**X**										
	[117]	**X**									**X**		**X**				
Rabbit	[20]	**X**				**X**											
	[93]	**X**				**X**											
	[28]					**X**	**X**	**X**									
	[30]					**X**	**X**	**X**									
	[118]					**X**											
	[25]	**X**				**X**											
	[119]					**X**											
	[120]					**X**											
	[121]					**X**											
	[122]				**X**												
Dog	[123]	**X**					**X**				**X**		**X**				**X**
Pig	[23]	**X**				**X**					**X**						
Sheep	[29]			**X**	**X**	**X**		**X**				**X**					
	[124]	**X**				**X**		**X**	**X**								
NHP	[50]	**X**				**X**				**X**							
	[21,22]	**X**														X (EMG)	**X**
	[44]	**X**															
	[56]	**X**	**X**			**X**					**X**			**X**		X (EEG, SSEP)	**X**
	[125]	**X**															
	[24]	**X**															**X**
	[126]	**X**				**X**	**X**		**X**	**X**							**X**
	[57]	**X**															
	[127]	**X**															
Human	[8]	**X**		**X**												X (EEG)	**X**
	[51]	**X**				**X**								**X**			**X**
	[54]	**X**												**X**			**X**
	[52]	**X**	**X**	**X**													**X**
	[9]	**X**															**X**
	[10]	**X**	**X**	**X**													**X**
	[11]	**X**															**X**
	[53]	X															X

***PET:***
*positron emission tomography;* **Δ*T:***
*thermometry; **H/E:** haematoxylin and eosin; **TUNEL:** terminal deoxynucleotidyl transferase dUTP nick end labelling; **VF:** vanadium acid fuchsin; **LB:** Luxol fast blue; **CV:** Cresyl violet; **PB:** Perl’s Prussian blue; **Iba1:** ionized calcium-binding adaptor molecule 1; **GFAP:** glial fibrillary acidic protein; **PCR:** polymerase chain reaction; **WB:** Western blotting; **ELISA:** enzyme-linked immunosorbent assay; **EMG:** electromyogram; **EEG:** electroencephalography; **SSEP:** somatosensory evoked potentials*

### 3.5. Parameters Influencing Ultrasound-Mediated BBB Disruption

After an extensive review of all protocols identified amongst included studies, the following parameter domains have been frequently investigated to assess their influence on the efficacy and safety of ultrasound-mediated BBB disruption: microbubbles, transducer frequency, peak negative pressure (PNP), pulsed delivery parameters (see Figure 5), the duration of each sonication, and the dosing of ultrasound application. The transducer frequency defines the frequency of the generated ultrasound wave, and the PNP reflects the amplitude or intensity of the wave. Detailed parameters from each included study are listed in Appendix A, and a summary of the influence of each parameter domain is listed in Table 3 and Table 4.

#### 3.5.1. Microbubbles

Five major commercially available microbubble formulations—Definity^®^/Luminity^®^, Optison^®^, SonoVue^®^/Lumason^®^, Sonazoid^®^, and Usphere^®^—have been utilised in studies investigating ultrasound-mediated BBB disruption. For reference, a comparison of these microbubble formulations, as well as their frequency of use and typical dosing in included studies, is included in Table 3. In addition, a handful of studies used in-house microbubbles, some of which were drug-loaded [38,45,64,94], in an attempt to further potentiate localised mechanical effects to move therapeutics across the BBB. Studies have also demonstrated the potential of BBB disruption without microbubble administration [65,89]; however, this was accomplished with a significant thermogenic effect [89] or by using higher intensity ultrasound waves [65]. Direct comparisons of microbubble administration against no administration have shown significant improvements in BBB disruption when microbubbles are administered, at unifying parameters [90]. Without microbubbles, markedly higher PNP sonications are required to achieve equitable BBB disruption, at the cost of poorer safety outcomes [32,65]. A previous review [17] commented on the complexities of assessing the effect of microbubbles on BBB disruption, referencing the lack of an accepted protocol for handling and administering microbubbles, as well as the intersubjective differences in cardiovascular function that result in variable microbubble concentrations at target locations. While most investigations employing commercially available microbubbles have cited adherence to manufacturer instructions on handling and preparation of microbubbles, pre-activation vial temperature [129] and time between decanting/administration [130] have been shown to alter the size distribution when using Definity^®^ microbubbles. As for the intersubject variability in cardiovascular function, this holds true for any administered agent that relies on the cardiovascular system for transport and accumulation in specific tissue vasculature and is a variable accounted for across the large number of studies included in this review. According to the review of the included literature, current in vivo evidence suggests the following microbubble-related factors influence the degree of BBB disruption and its safety: (1) microbubble characteristics, (2) dosing, and (3) timing/method of administration.

##### Microbubble Characteristics

A comparison of three commercially available microbubble formulations (SonoVue^®^ vs. Definity^®^ vs. Usphere^™^) in rats found comparable BBB opening and safety results between Definity^®^ and Upshere^™^, while sonications with SonoVue^®^ yielded significantly greater BBB opening than the other two microbubble formulations, at the lowest investigated ultrasound intensity [112]. More specific rodent investigations [85] sought to compare the effect of differing microbubble gas core composition (C_3_F- vs. C_4_F_10_- vs. SF_6_-filled), by administering equal doses of in-house microbubbles with identical shell composition and sizes. C_3_F_8_- and C_4_F_10_-filled microbubbles yielded significantly greater BBB disruption than SF_6_-filled ones, suggesting that microbubble gas composition specifically influences the ability to induce BBB disruption. These findings correlated with additional comparative findings where Sonazoid^®^ (C_4_F_10_-filled) microbubbles yielded significantly greater BBB disruption than comparably sized SonoVue^®^ (SF_6_) microbubbles, at unifying doses, administration timing, and ultrasound exposure parameters. The influence of microbubble size or diameter has also been investigated across three studies that directly compared compositionally identical, in-house microbubbles of different average diameters (1 to 2 µm vs. 4 to 5 µm [75]; 1 to 2 µm vs. 4 to 5 µm vs. 6 to 8 µm [66,72]; 2 µm vs. 6 µm [110]). Findings concluded that larger diameter microbubbles caused a linear increase in BBB disruption [72], resulting in more prolonged [66] BBB opening, while only mildly elevating the potential for tissue damage [72]. This trend was supported by another study [45] that compared SonoVue^®^ (2.5 µm) with in-house (1.1 µm) microbubbles of similar composition. Additionally, increased proinflammatory gene expression has been observed in one study with the use of larger (4.2 µm vs. 1 to 1.5 µm) microbubbles, albeit with differing gas compositions [105]. Based on these data, the administration of larger diameter microbubbles appears to reduce the threshold for achieving BBB opening, requiring lower PNP sonications. Ultimately, a range of microbubble formulations have been used for safe ultrasound-mediated BBB disruption with an appropriate selection of sonication parameters, but differences in the efficacy of each microbubble formulation do appear and are likely attributable to variations in microbubble characteristics between formulations. Thus far, only Definity^®^ [8,131] and SonoVue^®^ [9,54] microbubbles have been used in clinical trials, likely due to their FDA approval and frequent use in preclinical studies.

##### Microbubble Dosing

The association between microbubble dosing, BBB disruption, and subsequent safety outcomes has been studied in numerous investigations with rodent and rabbit subjects [19,26,33,34,40,62,63,76,79,110,113,120]. The consensus from these findings is that using escalating microbubble dose only mildly increases the disruption and opening of the BBB, an effect that is often statistically insignificant when quantified [19,60,120]. Additionally, there seems to be an upper threshold microbubble dose for which subsequent administration of larger doses seems to cause BBB disruption to plateau [132] or paradoxically decrease [33,34,79] in rodents. Aberrations in this trend were noted when a range of microbubble doses were investigated in combination with (1) non-pulsed, unfocused ultrasound from a diagnostic, imaging transducer [39]; and (2) pulsed ultrasound, using a focused transducer [63]. In these studies, a significant positive correlation between microbubble dose and degree of BBB disruption was established, albeit with extensive tissue damage at higher doses. From available data, we can conclude that escalating microbubble dose may yield mildly elevated BBB disruption, usually up to a certain upper threshold dose, and with some heterogeneity in this trend observed among a few of the identified investigations. On the other hand, there seems to be a more consistent relationship between escalating microbubble doses and the increased risk of adverse safety outcomes reported by these same studies. This includes numerous reports of significant tissue damage [33,34,79], increased expression of proinflammatory genes associated with the NF-kB pathway [26], and greater cellular apoptosis [33,34].

##### Timing and Method of Microbubble Administration

In all identified protocols, microbubbles were administered via an intravenous route and were usually administered immediately prior to or at the onset of ultrasound application. As per the method of administration, evidence [79,108] has suggested that a prolonged intravenous infusion across the entire sonication period yields more reproducible and consistent, but not necessarily greater, BBB disruption when multiple cerebral foci are targeted [126]. It is theorised these differences can be attributed to the rapidly changing intravascular concentration of microbubbles attributed to bolus dosing, versus more stable microbubble availability attributable to infusion dosing [17]. Conversely, one rodent study assessed the extent of microbubble administration over 30 and 180 s infusion periods, reporting no significant difference in the extent of BBB disruption [76]. There is also some evidence to support that an infusion administration may yield less oedematous foci on T2 MRI, as compared to bolus administration [108]. More recent clinical trials have adopted microbubble infusion protocols continuously throughout the applied sonications [131,133].

**Table 3 pharmaceutics-14-00833-t003:** Comparison of five major commercially available microbubble formulations used in studies for ultrasound-mediated BBB disruption (information sourced from manufacturer) and typical doses.

Agent	Manufacturer	Shell Composition	Gas Core Composition	Mean Bubble Diameter (µm)	Bubble Concentration (Bubbles/mL)	Use in Identified Studies
Definity^®^/Luminity^®^	Lantheus Medical Imaging	Lipid	C_3_F_8_	1.1–3.3	1.2 × 10^10^	Used in *n* = 42 preclinical studies (typical doses: 10–20 µL/kg) and *n* = 6 clinical studies (typical dose: 4 µL/kg)
Optison^®^	GE Healthcare	Protein	C_3_F_8_	3.0–4.5	5–8 × 10^8^	Used in *n* = 14 preclinical studies (typical doses: 50–100 µL/kg but significantly varied in mice studies)
SonoVue^®^/Lumason^®^	Bracco Diagnostics	Lipid	SF_6_	1.5–2.5	1.5–5.6 × 10^8^	Used in *n* = 29 preclinical studies (typical doses 25–150 µL/kg) and *n* = 2 clinical studies (typical dose: 100 µL/kg)
Usphere Prime^®^	Trust Bio-sonics	Lipid	C_3_F_8_	1.0	2.8 × 10^10^	Used in *n* = 1 preclinical study
Sonazoid^®^	GE Healthcare	Lipid	C_4_F_10_	2.0–3.0	9 × 10^8^	Used in *n* = 1 preclinical study

#### 3.5.2. Transducer Frequency

While a range of transducer frequencies, from 28 kHz [23,102] to 10 MHz [45], have been applied to disrupt and open the BBB, the majority of these protocols have employed frequencies that fall within a narrower range of 0.2 to 1.5 MHz amongst in vivo subjects. Among clinical trials, data currently exist for the application of only three ultrasound frequencies—ExAblate^®^ Neuro (0.22 MHz), SonoCloud-1^®^ (1.05 MHz), and NaviFUS^®^ (0.5 MHz). In general, lower frequency ultrasound application (e.g., 28 kHz) has been shown to have greater tissue penetration, but a wider tissue focus, resulting in less targeted, ill-defined BBB disruption [102]. Conversely, higher frequency ultrasound beams tend to be more collimated, less tissue penetrative [94], and more likely to undergo tissue attenuation, resulting in greater beam aberration [134] and thermal energy liberation in the surrounding tissue, especially at the bony interface of the skull. We identified four preclinical, parametric studies that directly investigated the effect of altering the central frequency of a single-element transducer on BBB disruption efficacy and safety outcomes [19,45,94,119].

*McDannold* et al. [119] evaluated sonications of a variety of frequencies (0.26 MHz [30,118] vs. 0.69 MHz [25,28,120] vs. 1.63 MHz [7] vs. 2.04 MHz [119]) from multiple rabbit investigations. In their comparison, escalating frequencies had a higher threshold for BBB opening, requiring more intense, higher PNP ultrasound to achieve BBB disruption. This led to the conclusion that the threshold for successful BBB disruption was more appropriately dependent on the mechanical index (MI)—a ratio of the PNP over the square root of the transducer frequency. An estimated MI of 0.46 was identified as a threshold at which successful BBB disruption was achieved across all tested frequencies. Following up on this work, subsequent investigations comparing sonications with 1 MHz vs. 10 MHz [45,94] and 0.5 MHz vs. 1.6 MHz [19] ultrasound transducers have been conducted in rats. Findings from these studies support the notion that significantly higher PNP sonications are required to achieve BBB disruption with escalating frequencies, further consolidating the idea of an MI-dependent threshold. However, an MI threshold for BBB disruption was not observed when comparing 1 MHz to 10 MHz sonications [45,94], as was observed between frequencies used in other studies [19,119], and this may be due to the larger difference in frequencies tested between these studies. Lower frequency sonications, both at an equivalent MI [45,94] and at equivalent PNPs [19], produced a much larger area of BBB disruption, accompanied by off-target involvement, upon gross evaluation of Evans blue extravasation. This is believed to be due to the production of standing waves [135], enhanced reflection of ultrasound waves at the skull, and re-penetration into the brain’s parenchyma [94].

Despite requiring greater PNPs to achieve BBB disruption, higher frequency sonications show mild to significantly favourable safety outcomes over lower frequency sonications, when observed 2–6 h after sonication. After applying higher frequency sonications, *McDannold* et al. [119] reported a subtle reduction in microhaemorrhagic damage in tissue sonicated with 2.04 MHz as compared to 0.26 MHz ultrasound but an increased density of these red blood cell extravasations relative to the area of tissue region exposed to the ultrasound. *Fan* et al. [45,94] highlighted significantly greater haemorrhagic and oedematous change on MRI, gross, and microscopic examinations in brains sonicated with frequencies of 1 MHz than 10 MHz, both when controlling for MI [94] and PNP [45]. It is important to note that *McDannold* et al. [119] *and Fan* et al. [45] both applied ultrasound to rabbits and rodents via a craniotomy site, thereby reducing the potential of beam defocussing and attenuation from transcranial application.

#### 3.5.3. Peak Negative Pressure

The effect of PNP has been directly examined in a plethora of studies, including in rodents [15,19,31,32,33,45,62,63,65,69,80,81,87,88,93,94,101,112], rabbits [20,25,28,30,118], sheep [29,124], and even clinical trials [9]. Unfortunately, the accurate determination of PNP remains challenging [18] as in vivo PNP is difficult to measure; instead, in vitro pressures are measured and combined with skull attenuation coefficients to provide an estimate of the in vivo PNP [31]. While this may impede comparisons of PNPs between studies, data and trends from within studies can be useful in determining optimal parameters for safe and effective BBB disruption. We found that sonication PNPs at which safe and effective BBB disruption has been accomplished ranged from 0.2 to 0.5 MPa in most preclinical animal studies (see Table 4), with some utilising higher PNPs safely with higher frequency transducers (>1 MHz) [32,45,94,103]. It has been difficult to establish a narrow range of PNPs routinely used amongst clinical trials, as in their design they each test a range of PNPs across repeated sonications, ranging from 0.48 to 1.15 MPa [9,54], and 2.5 [10] to 60 W [52] of applied power.

General findings suggest that a threshold PNP is required for a given frequency of applied ultrasound (threshold MI), after which BBB disruption is achieved [67,88]. There is then a narrow therapeutic window at which a positive dose–response relationship exists, where raising the applied PNP improves the degree of BBB disruption, without materially impacting safety [19,25,32]. After this, continued elevations in PNP confer improvements in BBB disruption and opening, but also worsen safety outcomes, achieving a state of dose-limiting toxicity [65,68,73]. Eventually, a plateau is obtained [32,63], at which further escalations in PNP cause insignificant improvements to BBB disruption, while continuing to worsen safety outcomes [25,32]. After surpassing the threshold PNP required for BBB opening, and prior to this relationship plateauing, preclinical studies have identified linear relationships with escalating PNP and the volume of BBB opening on MRI [66,72,75], the extravasation of dextrans [67], and the uptake of therapeutic agents such as Herceptin [31]. Clinical trials in glioblastoma patients have also shown an increasing degree of BBB disruption with escalating PNP sonications [9,54], albeit with increased incidence of oedema [9]. Interestingly, one study noted that the uptake of a chemotherapeutic agent, BCNU, increased as the sonication PNP was raised from 0.45 to 0.62 MPa, peaked at 0.62 MPa, but decreased with subsequent elevations in PNP (0.98 and 1.38 MPa), despite an increase in contrast enhancement on MRI at these higher pressures [101]. The PNP has also been shown to influence the size [69] and molecular weight [67] of agents capable of passing through the BBB, with current evidence suggesting that higher, and therefore less safe, PNPs are required for transporting larger substances. Additionally, sonications of increasing PNP have been shown to prolong the reversibility or closure time following the disruption and opening of the BBB [65,66]. Furthermore, one study reported the effect of BBB disruption produced with PNPs of 0.55 and 0.81 MPa on downregulating the immunohistochemical expression of a key cerebrovascular drug-efflux pump—P-glycoprotein—for 48 and 72 h after sonication, respectively [15]. This finding suggests that increasing the PNP of ultrasound may go beyond exerting mechanical effects on the BBB and may additionally cause biochemical changes that favour drug accumulation in the brain.

The increasing presence of adverse safety events associated with escalating PNPs has been proposed to be due to the increased frequency of inertial cavitation in microbubbles. These events are demonstrated by the presence of broad or wide-band acoustic emissions from sonicated microbubbles, detected via a receiving ultrasound transducer element [30,45,67,112]. Collectively, this monitoring process is known as passive cavitation detection or acoustic emissions monitoring and has resulted in a paradigm shift in sonication delivery, where a dynamic power ramp technique is utilised to determine optimal PNP as opposed to applying static PNP sonications. Here, power is incrementally escalated to produce sonications of graduating PNP, and this power is stabilised when ultraharmonic and subharmonic signals (indicating stable cavitation) are detected, or the power is reduced if any wide-band emissions (indicating inertial cavitation) are detected. This variable power ramp delivery protocol has been applied successfully to produce safe BBB disruption in rodent [26,93], NHP [50,126], and clinical studies [10].

#### 3.5.4. Pulse Characteristics

Amongst identified investigations, ultrasound is typically delivered in a non-continuous, pulsed manner, with a small minority applying a continuous ultrasound scheme [40,89,121,122]. A pulsed delivery approach has been adopted as a mainstay to limit the exposure time to ultrasound in delicate brain tissue and has been shown to significantly reduce the thermogenic effect [89] associated with the continuous application of ultrasound. Of the four studies that applied a continuous ultrasound paradigm, one study was able to disrupt the BBB reproducibly, without adverse histopathological events, using a diagnostic, imaging ultrasound transducer [40]. We identified two primary variables that have intimately influenced the efficacy and safety of ultrasound-mediated BBB disruption: the length or duration of each pulse, consisting of one or more excitatory cycles of acoustic pressure waves, and the pulse repetition frequency, how frequently these series of pulses repeat (see Figure 5). Additionally, more novel iterations in discontinuous ultrasound delivery have emerged, including the delivery of ultrasound bursts (consisting of shorter, phasic pulses) over more commonly used tonic pulses (consisting of a longer pulse) [59,60,83,108].

##### Pulse Length

While pulse lengths as low as 0.35 µs [39] and as high as 100 ms [19,23,102] have been used to disrupt the BBB via pulsed ultrasound, a majority of preclinical studies appear to use pulses of 10 ms in length. In clinical studies, pulse lengths of 2 to 3 ms [8,10,11,53] have been used with the ExAblate^®^ Neuro device, and pulse lengths of 10 ms [54] and 23.8 ms [9] have been used with the NaviFUS^®^ and SonoCloud-1^®^ devices, respectively. We identified six parametric, preclinical studies that directly investigated the efficacy and safety outcomes of a range of pulse lengths. Escalating pulse length from 0.1 to 10 ms appeared to consistently increase BBB disruption in all six studies; in two studies, subsequently higher pulse length appeared to yield no significant improvements in BBB disruption efficacy, whilst simultaneously worsening safety outcomes [7,76]. Three studies did not corroborate this plateauing effect with sonications of pulse length >10 ms, highlighting a larger area of BBB disruption following sonications of 50 [102] and 100 ms [19,102] pulses, respectively. This effect may be attributable to the use of a diagnostic, imaging transducer, operating at a lower central frequency (28 kHz), delivering higher MI (MI = 4.78) sonications in two of the three studies by *Liu* et al. [23,102]. However, the transducer type and parameters utilised by *Shin* et al. [19] were comparable to the two studies that did highlight a plateauing effect with lengthening pulses >10 ms, making this previously described trend [7,18,120] less definitive. The threshold PNP required to successfully open the BBB has been shown to decrease with escalating pulse lengths from 0.1 to 10 ms in one study [120], likely due to the greater cumulative effect from more prolonged ultrasound pulses. Additionally, with sonications of pulse lengths ≥10 ms, the spatial distribution of tracers appears more heterogeneous, with significantly greater accumulation around blood vessels and less even parenchymal distribution than is observed with pulse lengths <10 ms [76,83]. Nonetheless, from the currently available literature, it appears that sonications of pulse lengths ≤10 ms appear to provide the greatest efficacy and safety benefits for ultrasound-mediated BBB disruption, when controlling for all other parameters.

##### Phasic vs. Tonic Pulses

More recently, the use of rapid, short pulse sonications or phasic pulses consisting of bursts (as opposed to more continuous tonic pulses) has been investigated for its potential to disrupt and open the BBB more homogeneously. Benefits of phasic pulses are theorised to occur via increased intraburst microbubble transit time, and the reduction in standing waves afforded with shorter, phasic pulse sequences [60,108]. Reports of ultrasound-mediated BBB disruption with phasic pulses have explored the use of pulse lengths ranging from 2.3 to 5 µs in length. Direct comparisons between phasic and tonic pulse sequences in mice have had mixed results, with some studies [82,83] reporting safer BBB disruption, with improved homogeneity and improved BBB reversibility, and another [59] reporting worse safety outcomes, with no improvements in the homogeneity or reversibility of BBB disruption with phasic pulsed schemes. These conflicting findings may be attributable to the differences in ultrasound frequency (1 [83] vs. 1.78 MHz [59]), microbubble type, and administration method (30 s infusion [83] vs. bolus [59]) between these studies. All studies thus far have exhibited a lower degree of BBB disruption with phasic pulses at unifying parameters, suggesting this protocol may provide a more conservative opening of the BBB. The use of phasic pulse regimens could provide safer, better-distributed delivery of therapeutics into the CNS, but it currently requires further investigation across a broader set of parameters for more conclusive data.

##### Pulse Repetition Frequency

Most sonication protocols that we identified utilised pulse repetition frequencies that fell within a range of 1–10 Hz in preclinical and clinical studies that employed single-element ultrasound transducers [9,54]. Clinical trials utilising the ExAblate^®^ multi-element, phased array device appear to utilise pulse repetition frequency of 30 to 31 Hz instead. The effect of pulse repetition frequency on BBB disruption efficacy and safety has been studied in a limited fashion, by five parametric studies that investigated this relationship [19,60,76,108,120]. Evidence from these studies appears to suggest a threshold pulse repetition frequency, and therefore a minimum number of pulses over a given duration of sonication, required to achieve successful BBB disruption [76]. After surpassing a relatively low pulse repetition frequency threshold, the effect of escalating pulse repetition frequencies has been inconsistent. Two studies reported no statistically significant improvement in BBB disruption minutes following tonic pulsed sonications of pulse repetition frequency 1–25 Hz [76] and 0.5 to 5 Hz [120], respectively. Contrary to these findings, three studies have reported significantly improved BBB disruption with escalating pulse repetition frequencies, both with longer, tonic pulsed sonications (pulse repetition frequency 1 to 5 Hz) [19] and with shorter, phasic pulsed sonications as part of a burst sequence (pulse repetition frequencies 6250 to 100,000 Hz [60] and 1 to 166,666 Hz [108]). Safety outcomes from escalating pulse repetition frequencies were either mildly improved [108] or showed no significant differences [19,60,76,120] within hours following the last sonication. These inconsistencies warrant further parametric study into the effect of using higher pulse repetition frequency sonications and may help in further optimisation of safer parameters.

#### 3.5.5. Sonication Duration

The sonication duration is another parameter that tends to affect the efficacy and safety of ultrasound-mediated BBB disruption, as it describes the time of exposure to ultrasound in one given application. Amongst all the study protocols we reviewed, the majority of sonication durations fell between 0.5 and 2 min, and this remained consistent in the protocols of clinical studies as well. Sonication durations as low as 6 s [121,122], with non-pulsed ultrasound, and as high as 20 min [93], with pulsed ultrasound, have also been shown to induce sufficient BBB disruption. From parametric studies, there appears to be a positive correlation between increasing sonication duration and the degree of BBB disruption, with the eventual trade-off being worsening safety outcomes following exposure to excessively long sonications both with pulsed [19,20,102] and continuous delivery ultrasound-mediated BBB disruption [40,121]. Additional data suggest that eventually a threshold is reached, where the effect of increasing the sonication duration saturates [93,115], and excessive tissue damage is observed [93]. Interestingly, one investigation digressed from this trend, where significant changes in the sonication duration resulted in mild but statistically insignificant increases in BBB disruption, without any observed histopathological change [76]. The authors of this study hypothesised that this was due to the fact BBB disruption saturating potential had already been achieved using the lowest tested sonication duration of 30 s, and this may be attributed to the higher frequency of pulsed ultrasound used, as compared to the other parametric studies identified.

#### 3.5.6. Dosing (Number and Frequency) of Ultrasound Applications

In this review, we divided the application of ultrasound into two categories—a sonication and a session. A session was defined as an application period, consisting of one ultrasound sonication or numerous ultrasound sonications separated by an interval of usually minutes to an hour (intersonication interval). Sessions are usually separated by a larger interval of time, on the timescale of days to weeks apart (intersession interval). The partition of ultrasound applications into these categories helps in understanding the effect of cumulative ultrasound acutely (after multiple sonications) and chronically (after multiple sessions). Ultimately, multiple sonications, and multiple sessions of ultrasound-mediated BBB disruption over months to years, would need to be safely tolerated if this approach is to achieve widescale clinical use for improving drug delivery in patients with CNS malignancies, dementias, and other neurodegenerative diseases.

##### Ultrasound Sonications

Repeating an ultrasound sonication once (double sonication) has been shown to significantly improve the magnitude of BBB duration and duration of BBB opening when compared to a single sonication [71,115] at the same target site. Additionally, the choice of the intersonication interval may influence the penetration and uptake of drugs into the brain, and specific drug half-lives may need to be considered when determining the most appropriate intersonication interval [71]. Unfortunately, efficacy data from more than repeat sonications are non-existent, as no further studies have directly compared the effect of an equivalent single sonication against more than two repeat sonications. Other studies, both preclinical and clinical [9,52], have tested a greater range of repeat sonications but have not adequately reported group-specific data on BBB disruption efficacy. When directly compared, the safety of double sonications appears to be similar to [71] or slightly worse than [115] a single sonication. Indirect comparisons from two different studies that utilised equivalent ultrasound parameters highlighted worsening MRI and histological safety outcomes after increasing the number of sonications per session from two [48] to four [49], over three weekly ultrasound sessions. Safety outcomes from other protocols employing repeat sonications have been generally favourable, but also variable, with some studies reporting worsening outcomes [33,93] but most reporting no differences in healthy [84,109], aging [29], and glioma animal models [43,78] and clinical [8,11,54] studies. Additional evidence also suggests that multiple, lower PNP/MI sonications can produce a greater area of BBB disruption than a single, higher PNP/MI sonication, with improved safety outcomes [103].

##### Ultrasound Sessions

The effect of multiple, repeat ultrasound sessions on the degree of BBB disruption is unclear, as the few identified studies that directly compared single against multiple ultrasound session applications did not investigate differences in the efficacy of BBB disruption between these groups [43,93]. In addition, studies that conducted multiple, repeat sessions of ultrasound-mediated BBB disruption (Table 4) have not sought to investigate the potential of a cumulative effect of these sessions on the long-term integrity and permeability of the BBB. Subsequent ultrasound sessions have been shown to require sonications with gradually escalating PNPs in order to achieve BBB disruption in animal models [93,99,100]. This is likely attributable to the general growth and the increase in skull thickness observed in animal models [70,71,99] and has not been a finding corroborated in adult clinical trials [9,53]. Adverse radiographic safety outcomes following long-term, repeat ultrasound sessions have been generally favourable in NHP [70,84] and clinical studies [9,53]. However, transient MRI lesions (suggestive of microhaemorrhagic and oedematous change) have developed following repeat sessions in some NHP [21,22] and human [9,10] subjects. Long-term behavioural and clinical evaluations appear to be favourable in rodents [84], NHPs [21,22], and humans [52]. Histopathological investigation of these NHP and human studies has been limited, and only half of NHP [50,56,126] and no human studies investigated any histological outcomes following repeat sessions of ultrasound-mediated BBB disruption. Adverse histological outcomes in NHPs have ranged from minimal to moderate microhaemorrhagic change and, in one study [126], occurred despite the absence of any MRI abnormalities. Rodent investigations [99,100] have observed worsening adverse safety events after numerous, weekly ultrasound sessions. These include permanent structural changes on MRI (microhaemorrhagic/oedematous lesion, enlarged ventricles), histopathological evidence of macrophage infiltration, increasing accumulation of phosphorylated tau [100], and evidence of neurogenesis [100]. Furthermore, one study reported worsening tissue damage after multi-session ultrasound applications coupled with liposomal doxorubicin delivery in a glioma rodent model [70]. Interestingly, no tissue damage was observed after multiple ultrasound sessions without liposomal doxorubicin co-administration [70], suggesting that the repeat co-administration of certain chemotherapeutic drugs may either directly damage surrounding tissue or lower the threshold for inertial cavitation-induced ultrasound damage. These study findings, as well as the occurrence of a sparse number of transient MRI abnormalities in NHP and clinical studies with already limited sample sizes, highlight uncertainty around the chronic safety of ultrasound-mediated BBB disruption on tissue health, challenging the narrative that repeat sessions of ultrasound-mediated BBB disruption are generally safe, as presented in prior reviews [17,18].

## 4. Discussion

After an extensive systematic review of currently available literature, we feel we have comprehensively summarised the parameters used in published protocols of ultrasound-mediated BBB disruption for enhanced drug delivery, as well as the subsequent effects on efficacy and safety (Appendix A). We have also listed parameter ranges at which effective BBB disruption has been conducted with the most favourable safety outcomes (Table 4). The heterogeneity in protocols used to ultrasonically disrupt the BBB in included studies is apparent; thus, more rigorous standardisation is required, especially in the setting of clinical trials. In addition, we have identified several areas related to the procedure itself, as well as the techniques used to analyse its efficacy and safety, where the body of current knowledge is less established.

Firstly, in relation to the procedure of ultrasound-mediated BBB disruption, an emerging subset of investigations in this field have proposed the benefit of shorter, phased pulses of ultrasound over longer, tonic pulsed schemes that have predominated thus far. The use of phasic pulses may improve the homogeneity and safety of ultrasound-mediated BBB disruption, at the cost of opening the BBB to a smaller degree. As a result, this ultrasound pulse protocol may be applicable in frequent sessions of BBB disruption for the delivery of therapeutics for less aggressive, chronic CNS conditions, but further work is required to translate these findings beyond the subset of rodent studies currently available. Advances in microbubbles, namely in designing and testing microbubbles with more optimal characteristics (larger diameters and C_3_H_8_ or C_4_H_10_ gas filling) may also play a role in enhancing the efficacy and safety of ultrasound-mediated BBB disruption.

Secondly, the methods used to confirm ultrasound-mediated BBB disruption have relied upon the visualisation of proxy markers, mainly histological tracers or gadolinium-based MRI contrast agents. While these tracers have been essential in demonstrating proof of concept of ultrasound-mediated BBB disruption, they are ultimately not the intended therapeutic molecules needing to be delivered into the CNS. Studies have identified that the molecular weight, half-life, and timing of administration [70,71] influence the ability of a drug to traverse a disrupted BBB following ultrasound, and thus more research is required to track the uptake and transport of drug molecules not only across the BBB but to desired target cells.

Thirdly, the type of safety assessments conducted throughout the investigations we identified have overwhelmingly focused on structural alterations in sonicated neural tissue, both at gross and microscopic anatomical levels (e.g., haemorrhagic, cellular, and oedematous change). This has created a gap in our understanding of the physiological changes that follow ultrasound-mediated BBB disruption and, of note, the possibility of long-term inflammatory changes persisting after the passing of cerebrovascular contents through the BBB. Current proteomic and transcriptomic analyses seem to suggest an upregulation of proinflammatory genes following ultrasound-mediated BBB disruption, but the effect of this, if any, on neural tissue functioning remains to be seen. Electrophysiological changes following parameters used for ultrasound-mediated BBB disruption is another understudied area, particularly as neuromodulation and stimulation is an emerging area of therapeutic ultrasound research [128]. Surprisingly, none of the studies we identified sought to assess the impact of ultrasound delivery on the cranial bone and surrounding soft-tissue structures (skin, connective tissue, galea), even though most protocols involve ultrasound application transcranially, and the cranium remains the first point of tissue contact with the ultrasound beam.

After reviewing studies that repeatedly disrupted the BBB over chronic testing periods, we feel there is insufficient evidence to suggest that ultrasound can be frequently and chronically applied without exhibiting some degree of damage. NHP and human studies trialling chronic sessions of ultrasound-mediated BBB disruption have reported the presence of some adverse events, mainly transient MRI, and behavioural/clinical abnormalities. Conversely, rodent studies have highlighted permanent MRI and histological adverse changes from chronic exposure. While these differences in safety outcomes may be attributable to interstudy protocol variability, or anatomical differences between humans, NHPs, and rodents, a definitive suggestion of repeatably safe ultrasound-mediated BBB disruption is difficult to make given the limited and conflicting dataset. While some adverse events, whether transient or permanent, may be an acceptable risk when treating advanced CNS conditions, the prevalence and long-term impact of any adverse event on pre-existing neurological morbidity are currently not known in humans. Additionally, current NHP and clinical evidence is limited, both by small sample sizes (n<10 in most studies) and the sparsity of histological and biochemical safety analyses. Pharmacological strategies such as dexamethasone administration in an attempt to attenuate inflammatory response following ultrasound-mediated BBB disruption may also play an important role in the clinical adoption of this technique in treating chronic CNS conditions [106] and therefore represent another area of further research. Emerging evidence on the benefit of real-time imaging techniques such as Doppler [136] and photoacoustic imaging [136,137] may provide further technological advances in the clinical confirmation of ultrasound-mediated BBB opening without the need for MRI.

## 5. Conclusions

Greater standardisation of protocols and parameters used in preclinical and clinical studies investigating ultrasound-mediated BBB disruption is required for advancing clinical translation. Future studies should strive to further characterise the efficacy of ultrasound-mediated BBB disruption. This should focus on not only the opening of the BBB to MRI contrast agents, but also the delivery of intended drug molecules and their subsequent benefit in outcomes related to CNS conditions (e.g., reduced tumour progression and improved survival rates with high-grade cancers, reduced cognitive decline in dementia). Currently, numerous, larger clinical trials involving CNS cancer [138] (NCT04440358, NCT04528680, NCT04614493, NCT03744026, NCT04804709) and dementia (NCT04118764) patients are ongoing. The data from these trials will hopefully provide greater clarity to our overall understanding of the long-term safety, tolerability, and efficacy of cumulative ultrasound-mediated BBB disruption and enhanced drug delivery in patients with advanced CNS conditions.

**Table 4 pharmaceutics-14-00833-t004:** Summary of safe and effective parameters used in identified studies and reported relationships between parameter escalation and BBB disruption efficacy and safety outcomes.

Parameter	Safe and Effective Parameters Commonly Used	Parameters Compared	Reported Effects on BBBD (Efficacy Outcomes)	Reported Safety Outcomes
Transducer Frequency	Preclinical: 0.20–1.50 MHz Clinical: 0.22, 0.50, and 1.05 MHz	0.26, 0.69, 1.63, 2.04 MHz [119]	Increasing frequency: greater PNP required to achieve BBBD [19,45,94,119]; smaller foci/area of BBBD [19,45,94]	Increasing frequency: increased density of microhaemorrhagic activity [119]; decreased haemorrhagic [19,45,94] and oedematous activity [45]
		1 and 10 MHz [45,94]		
		0.5 and 1.6 MHz [19]		
PNP	Preclinical: 0.2–0.5 MPa with <1 MHz transducers Clinical: 0.48–1.15 MPa and 2.5–60 W power	0.30, 0.46, 0.61, 0.75, 0.98 MPa [73]	Increasing PNP: increasing BBBD after surpassing threshold PNP [9,15,45,65,73]; eventual saturation point in BBBD [32]; prolonged BBB opening [65,66]; prolonged P-glycoprotein downregulation [15]	Increasing PNP: increased haemorrhagic [15,19,65,66,73,93] and microhaemorrhagic change [73,93]; neuropil loss; neuronal loss [73,93] and necrosis [93]; evidence of apoptosis [45]; cerebral oedema [9,45]; hypoactivity/ataxia/tremor [33]
		0.55, 0.81 MPa [15]		
		0.27, 0.39, 0.59, 0.78 MPa [93]		
		0.3, 0.5, 1.0, 1.5, 2.0, 2.5, 4.5 MPa [45,94]		
		1.1, 1.9, 2.45, and 3.5 MPa [65]		
		0.45, 0.62, 0.98, 1.32 MPa [101]		
		0.55, 0.78, 1.1, 1.9, 2.45, 3.47, 4.9 MPa [32]		
		0.2, 0.3, 0.6, 1.5 MPa [19]		
		0.30, 0.51, 0.89 MPa [33]		
		0.4, 0.5 0.8, 1.1, 1.4, 2.3, 3.1 MPa [28]		
		0.2, 0.4, 0.5, 0.8, 1.1, 1.8 MPa [25]		
		0.78, 0.90, 1.03, 1.15 MPa [9]		
PL	Preclinical: 10 ms Clinical: 2–3, 10, and 23.6 ms	0.1, 0.2, 1.0, 2.0, 10, 20, 30 ms [76]	Increasing PL: increasing BBBD with PL 0.1–10 ms; statistically non-significant increase in BBBD after PL > 10 ms [7,76]; decreased PNP threshold (PL = 0.1–10 ms) [120]; heterogeneous distribution of BBBD/greater perivascular accumulation of tracer [76]	Increasing PL: no microhaemorrhagic change with PL ≤ 10 ms [19,120]; significant haemorrhagic change with PL = 100 ms [19,23]; evidence of apoptosis with PL = 100 ms [23]
		1, 10, 100 ms [19]		
		10, 100 ms [7]		
		0.1, 1, 10 ms [120]		
		30, 100 ms [23]		
		10, 50 and 100 ms [102]		
PRF	Preclinical: 1–10 Hz Clinical: 1–10 Hz and 30–31 Hz	0.1, 1, 1, 10, 25 Hz[76]	Increasing PRF: no BBBD with PRF = 0.1 Hz [76]; inconsistent improvements in BBBD with tonic pulsed sequences, some being statistically significant [19] and others not [76,120]; improvements in BBBD with rapid, phasic pulses [60,108]	Increasing PRF: no increase in adverse safety outcomes, via MRI [108] and histology [19,60,76,108,120]
		0.5, 1, 2, 5 Hz [120]		
		1, 2, 5 Hz [19]		
		1, 1667, 3333, 16,667, 166,667 Hz [108]		
		6250, 25,000, 100,000 Hz [60]		
SD	Preclinical: 30–120 sClinical: 30–120 s; 150–270 s in one study	30, 660 s [76]	Increasing SD: improved BBBD with pulsed [19,93,102] and continuously [40,121] applied US; plateauing effect thereafter [93,115]; one study reported no improvement in BBBD [76]	Increasing SD: minimal change in adverse safety outcomes with small increases, and significantly worsening safety outcomes with excessive increases [19,40,93,102,121]; no increase in histopathological outcomes in one study [76]
		240, 360, 480, 600 s [102]		
		30, 60, 120, 300 s [19]		
		30, 180, 300, 600, 1200 s [93]		
		60, 120, 180, 240 s[40]		
		6, 8 and 10 s [121]		
Dosing (Number and Frequency) of Sonications	Preclinical: 1–13 sonications/session (ISI = 5 min) Clinical: 1–8 sonications/session (ISI not stated) *ISIs are listed within brackets*	1, 2 (10 min), 2 (120 min) sonications [71]	Increasing sonication #: increase in BBBD [71,115]; improved doxorubicin uptake with shorter ISI [71]	Increasing sonication #: no [71] or mild [115] histopathological change (increased neuropil vacuolation)
		1, 2 (20 min), 2 (40 min) sonications [115]		
Dosing (Number and Frequency) of Sessions	Preclinical: 1–27 Clinical: 1–10 sessions *Intersession intervals are listed within brackets*	2–10, 2–6 sessions (biweekly and monthly) [84]	Increasing session #: higher PNP sonications required to achieve similar BBBD, but likely due to animal model growth [84] as not observed in developed adult clinical trials [9,11]	Increasing session #: no adverse safety outcomes [53,123]; transient MRI changes [9,21,22]; cortical atrophy, ventricular dilation, and lesion formation on MRI [100]; increased phosphorylated tau deposition [100]; increased neurogenesis [100] no change in motor and behavioural outcomes in rodents [84]; increased tissue damage and macrophage infiltration with doxorubicin co-delivery [70,91]; increasing number of apoptotic cells (significantly larger microbubble dose) [33]; mild increase in white matter vacuolation and mild neuronal injury (significantly larger microbubble dose) [43]
		1, 8 (3 days) sessions [43]		
		1, 4 (weekly) sessions [123]		
		1, 6 (weekly) sessions [100]		
		1, 3 (weekly) sessions [93]		
		1, 3 (weekly) sessions [70,91]		
		1, 2 (2 days), 3 (2 days) sessions [33]		
		3 (monthly), 6 (monthly) sessions [53]		
		4–27 (varying intersession intervals) sessions [21,22]		
		1–10 (monthly) sessions [9]		

***US:***
*ultrasound; **BBBD:** blood–brain barrier disruption; **PL:** pulse length; **PRF:** pulse repetition frequency; **SD**: sonication duration; **ISI:** intersonication interval.*

## Figures and Tables

**Figure 1 pharmaceutics-14-00833-f001:**
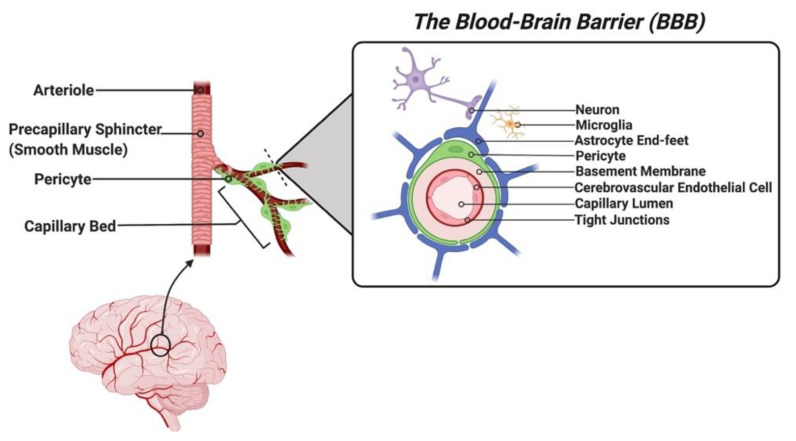
Schematic representation of the anatomical structure of the blood–brain barrier (BBB) and the accompanying neurovascular unit. Note the abluminal CEC surface is ensheathed by a basement membrane embedded with pericytes. (*Created with Biorender.com* (accessed on 12 March 2022)).

**Figure 2 pharmaceutics-14-00833-f002:**
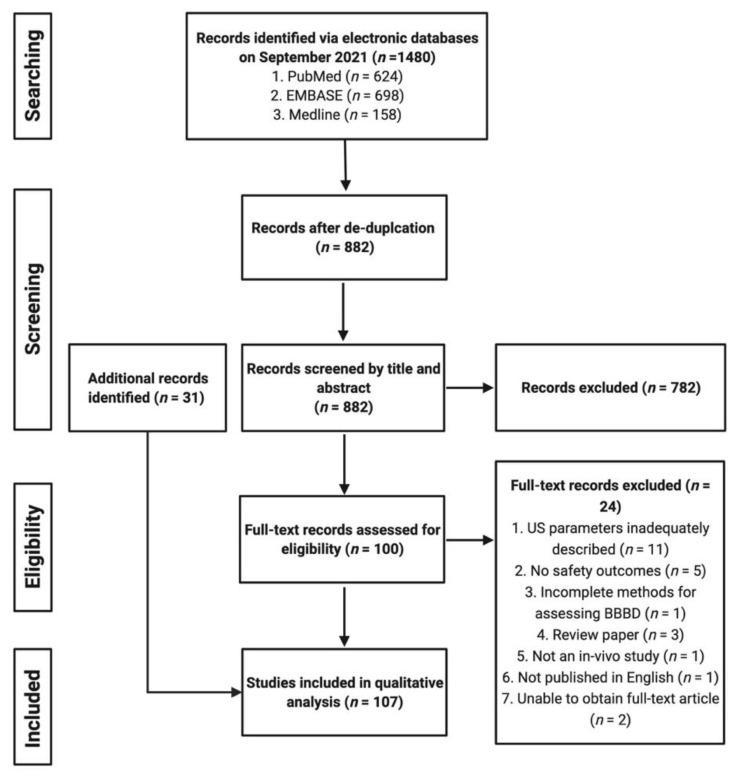
Flowchart highlighting the screening and selection process for studies included within this systematic review. (*Created with Biorender.com* (accessed on 12 March 2022)).

**Figure 3 pharmaceutics-14-00833-f003:**
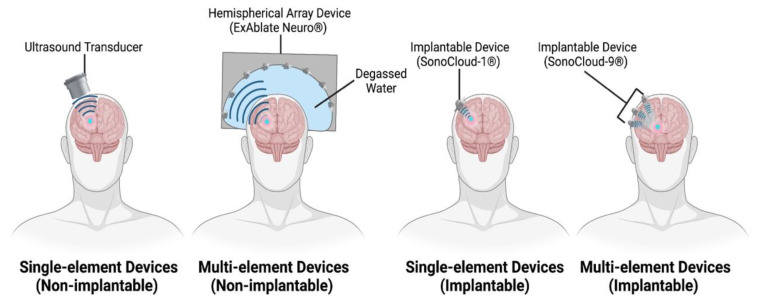
Comparison of therapeutic ultrasound devices used for BBB disruption. (*Created with Biorender.com* (accessed on 12 March 2022)).

**Figure 4 pharmaceutics-14-00833-f004:**
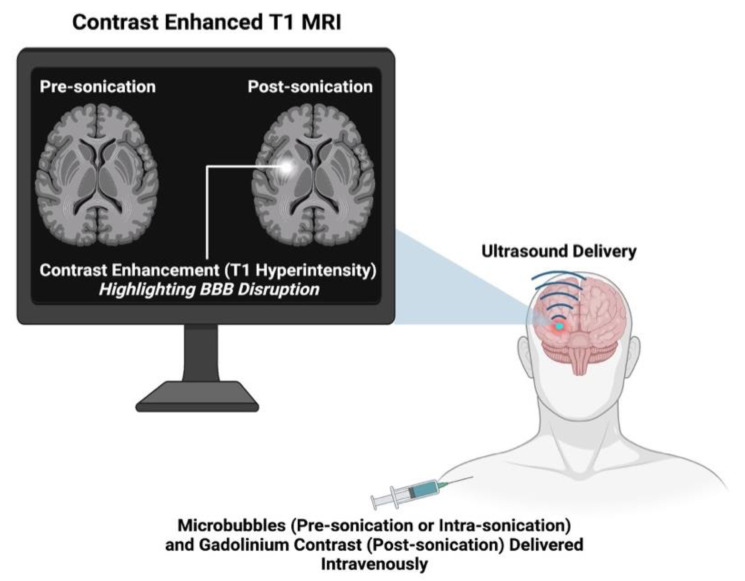
Representative cartoon highlighting the assessment and confirmation of ultrasound-mediated BBB disruption using CE-T1 MRI. (*Created with Biorender.com* (accessed on 12 March 2022)).

**Figure 5 pharmaceutics-14-00833-f005:**
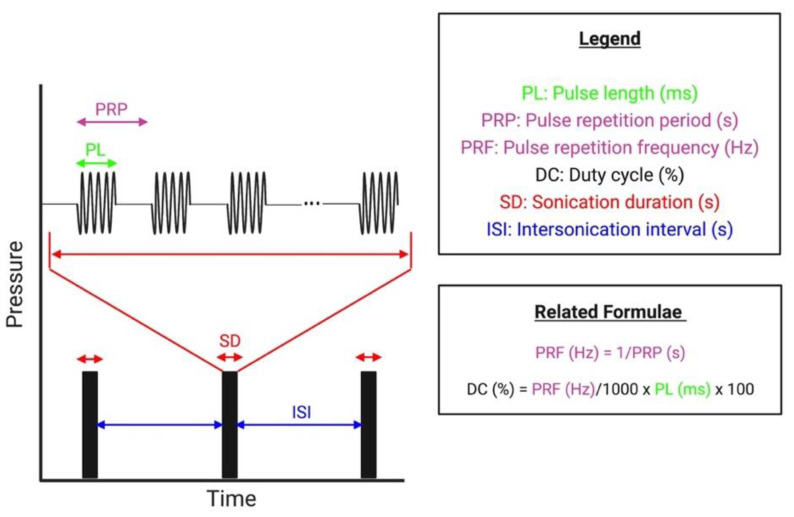
Overview of the pulsed delivery paradigm used for US delivery. (*Created with Biorender.com* (accessed on 12 March 2022)).

## Data Availability

All data supporting reported results can be found in the manuscript and supplementary materials of the manuscript.

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
