# Peer review of "Ultrasound-Mediated Blood–Brain Barrier Disruption for Drug Delivery: A Systematic Review of Protocols, Efficacy, and Safety Outcomes from Preclinical and Clinical Studies"

_pharmaceutics, 2022, doi:10.3390/pharmaceutics14040833_

Round 1

Reviewer 1 Report

The authors provide a comprehensive summary of the protocols and technical setups in ultrasound-mediated BBB disruption, which helps standardize studies in this field and advance this technology. I suggest accepting this manuscript for publication after the following improvements:
(1) In line 247, please double-check the unit of “28 Hz”.
(2) I suggest the authors have a separate section to discuss the skull-induced aberration and attenuation to the acoustic field.
(3) I suggest that the authors have a separate section to highlight the clinical trials (separate from the preclinical studies).
(4) It would be more impressive to include some representative images (MRI or other modalities) in the assessment of the BBB disruption section.
(5) In addition to the MRI, how could newly immerged imaging technologies (e.g., ultrafast doppler, photoacoustic imaging) contribute to this field?
(6) This review article is comprehensive from a technical perspective. However, clinicians or other users may get lost in the technical details, and it would be helpful to summarize the unique advantages and examples in specific applications.
(7) Outlook for the field is essential in a review article. Please expand your conclusions to include more perspectives on the development of this technology.

Reviewer 2 Report

In this systematic Review the authors included 107 articles about ultrasound mediated blood-brain barrier (BBB) disruption. They aimed to summarise protocols/ ultrasound parameters, efficacy and safety of the procedure. They introduce the BBB and highlight the benefits and disadvantages of it for therapies. Furthermore, it is clear described that a reversible disruption of the BBB is necessary.

The presented systematic review was conducted according to the PRISMA (Preferred Reporting Items for Systematic Reviews and Meta-analyses) guidelines and not only based on a Pubmed search. Studies with the goal for cellular, viral or gene delivery, neuromodulation, stimulation, or tissue ablation were excluded from the review. Whole procedure of the review process is well described in the Material & Methods section. The authors explain and discuss almost all important issues regarding the application of ultrasound for BBB opening starting from the technical ultrasound systems, results of used bubble formations, frequency and e.g. sonication durations through to safety and changes in behavior in preclinical and clinical studies.

Overall, it is a very well structured review with great relevance for scientific world. A lot of effort has been made to present the review process as well as the results in a clearly structured way.

Abstract

The title indicate that authors reviewed articles from preclinical (phantoms? animals?) side and clinical applications but this information is missing in the abstract and need to be included. Sentences like „Information was extracted, collated, analysed, and summarised.“ in line 21 can be deleted since they give no real information and are repeated some time (line 26). The last sentence of the abstract „. Further investigation into the long-term safety and feasibility of this therapeutic approach is required in future studies.“is not identifiable as the consequence of the whole review process.

Minor issues:

Line 19: „to identify“ is missing

From line 21 there is drop in the text format.

Introduction

Overall, it is only critical to note that the authors always talk about ultrasound, although in most cases focused ultrasound was used e.g. line 92 reference 17. For better understanding focused ultrasound should be distinguished from e.g. imaging ultrasound. In principle a chapter describing ultrasound principle and techniques would complete the introduction or give some more details about US in 1.2.

Results

For a better understanding and to break up the mass of written text I would recommend to include images of the ultrasound systems (single element vs. Phased array; helmet vs. Implantat) as well as anatomical/MR images showing the opening of the BBB.

In Table 3 information about the optimal Bubble concentration could be added to make it easier for other researchers.

General things:

Reference style: The reference style for MDPI should not contain high-set numbers.-> [1]is used should be [1]. Found some space errors e.g. line 37 „collectivelyreferred“, line 241

Please verify uniform spelling e.g. re-positioning (line 289) vs. repositioned (line 291).

Explanation of the abbrevation NHP (non-human primates (NHPs)) is missing in the description of Table 1. Line 859 starts with small letter „. sonication“ and need correction
